# Eprinomectin and Moxidectin Resistance of Trichostrongyloids on a Goat Farm in Austria

**DOI:** 10.3390/pathogens11050498

**Published:** 2022-04-21

**Authors:** Barbara Hinney, Sandra Wiedermann, Waltraud Kaiser, Jürgen Krücken, Anja Joachim

**Affiliations:** 1Department of Pathobiology, Institute of Parasitology, Vetmeduni Vienna, Veterinärplatz 1, 1210 Vienna, Austria; sandra.wiedermann@vetmeduni.ac.at (S.W.); anja.joachim@vetmeduni.ac.at (A.J.); 2Tierärztliche Praxisgemeinschaft Passail OG, 8162 Passail, Austria; praxis@almenlandtierarzt.at; 3Institute for Parasitology and Tropical Veterinary Medicine, Freie Universität Berlin, 14163 Berlin, Germany; juergen.kruecken@fu-berlin.de

**Keywords:** anthelmintic resistance, macrocyclic lactones, *Haemonchus contortus*, *Trichostrongylus*

## Abstract

Anthelmintic resistance (AR) of trichostrongyloids is widespread in Europe, but there is no up-to-date information on the spread of AR in caprine parasites in Austria. Eprinomectin (EPR) is currently the only anthelmintic drug of the macrocyclic lactones registered for goats in Europe. The aim of the present study was to gather information regarding the efficacy of anthelmintics against trichostrongyloids on a dairy goat farm in Austria with reported treatment failure of macrocyclic lactones and to determine the presence of different trichostrongyloid genera. Faecal egg count reduction tests (FECRT) using Mini-FLOTAC were performed with eprinomectin (EPR) and moxidectin (MOX). Egg count reduction, calculated with the R package egg-Counts, was 44% for EPR and 86% for MOX, confirming AR of trichostrongyloids for both compounds. The most frequently detected genus in larval cultures was *Haemonchus,* followed by *Trichostrongylus*. This is the first report of MOX resistance in caprine trichostrongyloids in Europe. Failure of EPR and MOX to control trichostrongyloid infections is a severe threat to dairy goat farming, since other compounds must not be applied in goats used for milk production. *Haemonchus contortus* is one of the most pathogenic parasites of small ruminants and can quickly develop AR. Thus, immediate action should be taken to slow the further spread of AR in this and other roundworm species of ruminants in Austria.

## 1. Introduction

The development of anthelmintic resistance (AR) in trichostrongyloids of ruminants is a serious threat to the health and productivity of livestock. In Austrian sheep, we can already observe a dramatic situation, with the failure of all anthelmintic classes available for this species to control trichostrongyloid infections [1].

A recent meta-analysis about AR in Europe demonstrated that AR is widespread, but that there are also clear data gaps [2]. Especially regarding goats in Austria, more data on AR is considered to be essential [2]. Although goats are a minor livestock species in Austria, their popularity is increasing, and the number of goats has doubled during the last 20 years with 100,601 goats registered in 2021 (Statistik Austria 2021; http://www.statistik.at; Viehbestand, accessed on 22 February 2022).

Eprinomectin (EPR) is licensed for the treatment of goats against gastrointestinal roundworms and large lungworms as a pour-on (licensed since 2018 and on sale since 2020; Phillip Kukla, Boehringer Ingelheim, pers. comm.) and injectable product (licensed since 2021) (Austrian Federal Office for Safety in Health Care; https://aspregister.basg.gv.at, accessed on 22 February 2022).

Since no anthelmintic compound was licensed for goats prior to the registration of eprinomectin, drugs licensed for sheep or cattle had to be applied off-label. Due to the generally limited availability of anthelmintic drugs licensed for goats, this also applies to cases of insufficient efficacy of EPR, where other anthelmintic drugs from the macrocyclic lactones (ML) or benzimidazoles (BZ) are to be used in dairy goats. Due to different pharmacokinetic profiles, goats require considerably higher doses than other ruminants (e.g., eprinomectin pour-on application is recommended in a dose of 0.5 mg/kg of body weight for cattle and 1 mg/kg of body weight for goats [3]); consequently, underdosing is considered to be common (especially if drugs are used off-label) and considered as a driving factor of AR in trichostrongyloids of small ruminants [4].

Not only because of possible frequent and long-term underdosing, but also because a higher treatment frequency is necessary in goats due to a weak development of endogenous defence against gastrointestinal nematodes even in older animals, goats are suspected to particularly promote the development of anthelmintic resistance in small ruminants [4]. Pasturing sheep together with goats is therefore considered to be a risk factor for AR development in sheep. Accordingly, our observation of a high level of anthelmintic resistance in ovine trichostrongyloids in Austria [1] indicates that AR in caprine parasites could be a considerable yet unreported problem.

The aim of our study was to determine the occurrence of AR in trichostrongyloid species on a dairy goat farm in Austria with a history of treatment failure. As EPR is registered for the treatment of goats with zero milk-withdrawal time and is therefore commonly used in dairy goat farms, we especially aimed to monitor the efficacy of this compound.

## 2. Results

At the first faecal egg count reduction test (FECRT) with EPR, egg count reduction (ECR) was 44% (95% CI = 24–65; Figure 1). At the second FECRT with MOX, the FEC was 86% (95% CI = 75–93; Figure 1). Faecal egg count data are listed in Appendix A.

Before and after treatment with moxidectin, *Haemonchus* (Figure 2) was the predominant species in larval culture representing 79.8% and 79.3% of the larvae, respectively. The second most frequent species was *Trichostrongylus* with 11.5% and 6.9%, respectively. Other detected species with relative frequencies <10% were *Teladorsagia*, *Oesophagostomum* and *Cooperia*.

## 3. Discussion

According to the current WAAVP guidelines [5], AR is present when egg count reduction is below 95% and the lower CI is below 90%. Thus, for both drugs, AR was unequivocally observed. This is, to the authors’ knowledge, the first published report of MOX resistance in goats in Europe. However, unpublished reports of MOX failure in trichostrongyloids from Italy were mentioned in a meta-analysis [2]. Treatment errors can be ruled out, as animals were weighed before treatment and the full dose was administered by a veterinarian.

Since the licensing of ERP for goats and its marketing starting in 2002, EPR is the only licensed anthelmintic product for goats in Austria (and the only macrocyclic lactone in the European Union), and no other product can legally be applied to treat nematode infections in goats unless AR to EPR is demonstrated. Studies of the EPR pour-on formulation on goats showed good efficacy of this formulation in susceptible trichostrongyloid isolates in experimental infections [3,6]. However, some authors raised the concern that the pour-on formulation and recommended dose of EPR might be suboptimal due its highly variable bioavailability [7]. As cross-resistance of compounds from the same class of substances is well described [8,9], it is very likely that EPR, although comparatively recently released for goats, may not be efficient on goat farms where ML resistance is already present due to long-term off-label application of eprinomectin, along with other MLs, such as ivermectin or moxidectin. In line with that, high prevalence of EPR resistance in goats in Europe have already been reported in four neighbouring countries of Austria (Switzerland, Italy, Germany and Slovakia [10,11,12]). Therefore, EPR efficacy should be immediately monitored, even when applied for the first time on a farm.

The observation of MOX resistance in our study indicates the progressed stage of a more generalized ML resistance on the investigated goat farm [9]. Since BZ resistance has previously been described in Austria in sheep trichostrongyloids [1,13], it is of the utmost importance to tightly monitor the development of AR in the field and to monitor changes in the use of anthelmintics and the effects of such changes in order to not maintain single- or multi-resistant populations under selection pressure, since AR is considered to be non-reversible [14].

The results laid out above demonstrate, at least for the examined farm, a very critical situation. Resistances of trichostrongyloids against MOX, EPR and possibly BZ are a significant threat to the dairy goat production in Austria, as there are few alternative treatment options since monepantel and levamisole must not be used in dairy goats. This situation is especially threatening for organic farms, because, in this management type, pasturing of ruminants is mandatory in the EU (European organic farming regulation (EU 2018/848)) and long-term stabling of animals that suffer from GIT parasite infections would thus not be an option.

*Haemonchus contortus* is the most pathogenic trichostrongyloid species in ruminants and is known to develop AR particularly quickly [15]. As *Haemonchus* was also the predominant species on the examined goat farm, this is worrying, as anthelmintic failure might directly lead to severe health problems of animals.

In the present case, it is not clear how the detected AR developed on the examined goat farm. While long-term unmonitored treatment with the same or a small number of different compounds are driving factors for AR development, it is also possible that resistant worms are introduced to a farm unintentionally with new stock, and can pose a significant threat to the resident host population [16].

In conclusion, the observations of this study ask for immediate actions. First, veterinarians and goat farmers should be informed about the current situation, and the monitoring of anthelmintic efficacy on each goat farm is strongly recommended. Second, the implementation of alternative treatment control options and measurements to slow the further progression of AR should be employed. Amongst these measures are pasture management, quarantining of newly introduced stock, selective treatment approaches and the use of plant nutraceuticals as alternatives to synthetic anthelmintic drugs [4,17].

## 4. Materials and Methods

A dairy goat farm in Styria, Austria, with 29 goats and a history of anthelmintic treatment failure suspected by the attending veterinarian was examined with two faecal egg count reduction tests (FECRT), first with eprinomectin (EPN) and then with moxidectin (MOX). Information on goat breed and age can be found in Appendix A. At the first FECRT in October 2019, faecal samples of all 29 goats were taken rectally and examined by Mini-FLOTAC applying a multiplication factor of 5 according to Cringoli et al. [18]. More details are given in Untersweg et al. [1] (Figure 3). On the same day, all animals were weighed and treated with EPN (Eprinex^®^ pour on; Boehringer Ingelheim, Lyon, France; 1 mg/kg BW; since the licensed EPN pour-on formulation for goats was not available, the compound for cattle was used) by the attending veterinarian (W.K.). Two weeks after treatment, faecal samples of animals that had an EpG of at least 200 before treatment (14 goats) as well as 1 goat that was anaemic with a pre-treatment EpG of 105, were again examined by Mini-FLOTAC. Individual efficacy was calculated using the web interface (https://www.math.uzh.ch/as/index.php?id=software_as00 (last time assessed on 4 March 2022) based on the R package eggCounts 2.3 applying the “paired samples with individual efficacy” option, as described previously [19].

The second FECRT was performed with 0.4 mg/kg p.o. MOX (Cydectin^®^, Zoetis Österreich GmbH, Vienna, Austria) in May 2020 with 17 goats with an EpG of at least 200. At this second FECRT, larval cultures of pooled faecal samples pre- and post-treatment were performed by pooling all samples positive for strongyle eggs. Faeces were mixed with water and vermiculite and incubated at 25 °C for 13 days. On day 14, the third-stage larvae were harvested and identified (≈100 larvae per coproculture) using the identification key of van Wyk et al. [20].

## Figures and Tables

**Figure 1 pathogens-11-00498-f001:**
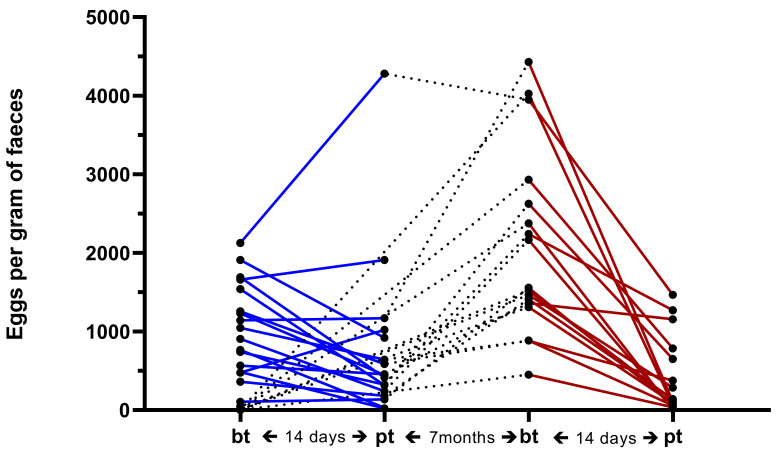
Individual egg per gram of faeces counts before (bt) and two weeks after treatment (pt) with eprinomectin (blue lines) and moxidectin (red lines). Dotted lines: connect last egg count results for individual goats with egg counts before moxidectin treatment.

**Figure 2 pathogens-11-00498-f002:**
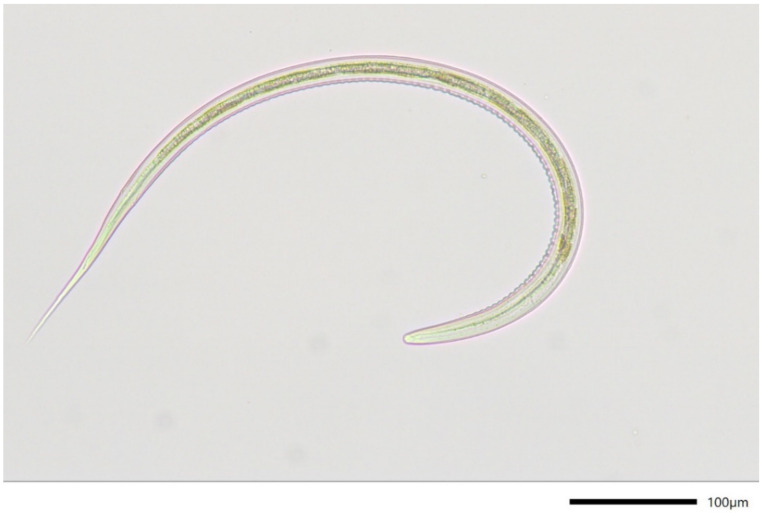
Third-stage larva of *Haemonchus contortus.*

**Figure 3 pathogens-11-00498-f003:**
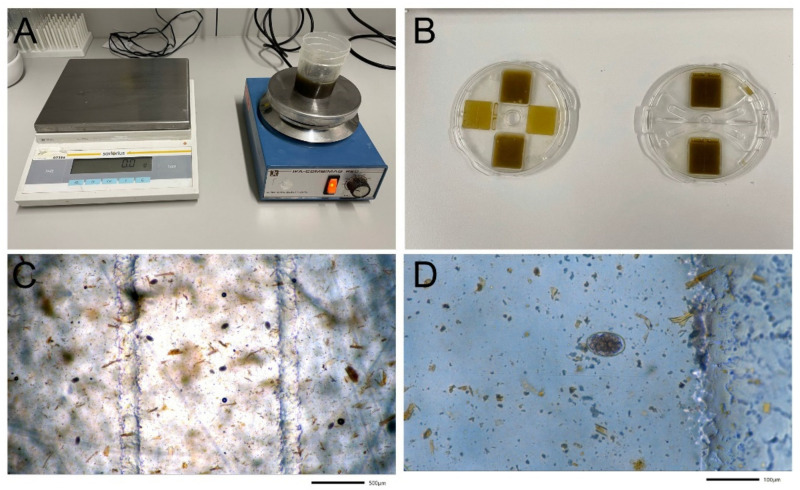
Overview on the method to quantify trichostrongyloid eggs. (**A**) The suspension of faeces and flotation medium is continuously mixed on the magnetic stirrer when the volume taken for loading the chambers is withdrawn to guarantee an even suspension; (**B**) 10 min after loading the chamber (**right**), the reading disc is turned for subsequent examination under the microscope (**left**); (**C**) trichostrongyloid eggs in the counting area (40× magnification); (**D**) 200× magnification.

## Data Availability

All data are available from the corresponding author upon reasonable request.

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
