# Peer review of "Eprinomectin and Moxidectin Resistance of Trichostrongyloids on a Goat Farm in Austria"

_pathogens, 2022, doi:10.3390/pathogens11050498_

Round 1

Reviewer 1 Report

In this paper, the authors address the phenomenon of anthelmintic resistance in the goats, a
species in which data is very lacking. This phenomenon is particularly serious and growing and
represents one of the main threats to the breeding of small ruminants, so this paper should be
considered for the Pathogens journal.
The manuscript is well written, however, I recommend a few minor corrections:
Line 17: Please indicates the quantitative technique used for the FEC.
Line 18: Please indicates the formula used to evaluate the FECR.
Lines 20 – 21: After the genus indicate the species or write spp.
Line 71: To make your results clearer, insert a table with the epg at D0 and the epg of the days in
which the FECR was assessed, also indicating the latter.
Line 77: After the genus indicate the species.
Line 77: After the genus indicate the species.
Lines 132-134: Also consider the use of natural products as a valid alternative to the synthetic
anthelmintic drug, for this reason I advise yo to read and refer to these recent manuscripts on
green veternary pharmacology:
Fabio Castagna, Cristian Piras, Ernesto Palma, Vincenzo Musolino, Carmine Lupia, Antonio Bosco,
Laura Rinaldi, Giuseppe Cringoli , Vincenzo Musella , and Domenico Britti. Green Veterinary
Pharmacology applied to parasite Control: evaluation of Punica granatum, Artemisia campestris,
Salix caprea aqueous macerates against gastrointestinal nematodes of sheep. Vet. Sci. 2021, 8,
237. https://doi.org/10.3390/vetsci8100237
Filip Štrbac , Antonio Bosco, Maria Paola Maurelli, Radomir Ratajac , Dragica Stojanovi ́c, Nataša
Simin , Dejan Orˇci ́c , Ivan Puši ́c, Slobodan Krnjaji ́c , Smaragda Sotiraki, Giorgio Saralli, Giuseppe
Cringoli and Laura Rinaldi. Anthelmintic properties of essential oils to control gastrointestinal
nematodes in sheep—in vitro and in vivo studies. Vet. Sci. 2022, 9, 93. https://doi.org/10.3390/
vetsci9020093
Line 141: After Miniflotac insert the correct reference:
Giuseppe Cringoli, Maria P Maurelli, Bruno Levecke, Antonio Bosco, Jozef Vercruysse, Jürg
Utzinger, Laura Rinaldi. The Mini-FLOTAC technique for the diagnosis of helminth and protozoan
infections in humans and animals. Nat Protoc. 2017 Sep;12(9):1723-1732. doi: 10.1038/nprot.
2017.067.
Supplementary data
In the table indicate, for each goat, also the epg at D0.

Author Response

Reviewer (R1): In this paper, the authors address the phenomenon of anthelmintic resistance in the goats, a species in which data is very lacking. This phenomenon is particularly serious and growing and represents one of the main threats to the breeding of small ruminants, so this paper should be considered for the Pathogens journal. The manuscript is well written, however, I recommend a few minor corrections:

Author (A): The authors want to thank the reviewers for their helpful revisions and comments to improve our manuscript.

  1. R1: Line 17: Please indicates the quantitative technique used for the FEC.

A: done (Line 18)

  1. R1: Line 18: Please indicates the formula used to evaluate the FECR.

A: We inserted the model used in line 19. The whole formula cannot be presented in a length that would be appropriate for the journal “Pathogens”. Since this relies on a very complex Bayesian model addressing different hierarchical levels of variation and the final estimate for the fecal egg count reduction is obtained by sampling from two Monte-Carlo-Markov chains, there is no simple formula to present. Formulas for the models are given in the open access paper of Wang et al. in section 2.2.2. https://www.sciencedirect.com/science/article/pii/S2211320718300599 which is also cited in our manuscript.

  1. R1: Lines 20 – 21: After the genus indicate the species or write spp.

A: With the morphological differentiation we applied it is not possible to differentiate the larvae up to the species level. Therefore, we would like to name the helminths on the genus level.

  1. Line 71: To make your results clearer, insert a table with the epg at D0 and the epg of the days in which the FECR was assessed, also indicating the latter.

A: These data are now included in the supplementary table

  1. Line 77: After the genus indicate the species.

A: see comment on Nr. 3

  1. Line 77: After the genus indicate the species.

A: see comment on Nr. 3

  1. Lines 132-134: Also consider the use of natural products as a valid alternative to the synthetic anthelmintic drug, for this reason I advise yo to read and refer to these recent manuscripts on green veternary pharmacology: Fabio Castagna, Cristian Piras, Ernesto Palma, Vincenzo Musolino, Carmine Lupia, Antonio Bosco,Laura Rinaldi, Giuseppe Cringoli , Vincenzo Musella , and Domenico Britti. Green Veterinary Pharmacology applied to parasite Control: evaluation of Punica granatum, Artemisia campestris, Salix caprea aqueous macerates against gastrointestinal nematodes of sheep. Vet. Sci. 2021, 8,237. https://doi.org/10.3390/vetsci8100237

Filip Štrbac , Antonio Bosco, Maria Paola Maurelli, Radomir Ratajac , Dragica Stojanovi ́c, Nataša Simin , Dejan Orˇci ́c , Ivan Puši ́c, Slobodan Krnjaji ́c , Smaragda Sotiraki, Giorgio Saralli, Giuseppe Cringoli and Laura Rinaldi. Anthelmintic properties of essential oils to control gastrointestinal nematodes in sheep—in vitro and in vivo studies. Vet. Sci. 2022, 9, 93. https://doi.org/10.3390/vetsci9020093

Thank you for these suggestions and the interesting references on this important field. We now added the strategy to use herbal products with anthelmintic efficacy in the sentence in line 136). However, the general strategy to use plant nutraceuticals is already mentioned in the review article on strategies to control goat parasites. Therefore, we only included one of the suggested references on sheep parasites (Line 136).

  1. Line 141: After Miniflotac insert the correct reference:Giuseppe Cringoli, Maria P Maurelli, Bruno Levecke, Antonio Bosco, Jozef Vercruysse, Jürg Utzinger, Laura Rinaldi. The Mini-FLOTAC technique for the diagnosis of helminth and protozoan infections in humans and animals. Nat Protoc. 2017 Sep;12(9):1723-1732. doi: 10.1038/nprot.2017.067.

A: Thank you! The correct reference is given now (line 143)

  1. Supplementary data In the table indicate, for each goat, also the epg at D0.

A: this is added now

Reviewer 2 Report

The article contributes to the field of veterinary helminthology by suggesting a new methodology for treating goats using different drugs and the association between them. I suggest that, based on the structure of the article and few results, this work could be presented as a short communication.

Small suggestions:

Introduction:
Page 2 (line 45-46): I suggest including a reference in this sentence.

Result:
To improve results in other trials of this project, I suggest including two new control groups. Using a similar treatment strategy, but repeating eprinomectin (twice) and moxidectin (twice). However, I think this is no reason to reject this article, with a short communication format.

Methodology:
Add the age of treated animals

Author Response

The article contributes to the field of veterinary helminthology by suggesting a new methodology for treating goats using different drugs and the association between them. I suggest that, based on the structure of the article and few results, this work could be presented as a short communication.

A: The authors want to thank the reviewers for their helpful and favorable review and the suggestion for improving our study

Small suggestions:

Introduction:

  1. R2: Page 2 (line 45-46): I suggest including a reference in this sentence.

Comment: Thank you for this comment. This sentence describes the situation in Austria, where no scientific publication is available for reference. However, in line 53 the general problem of off-label use in goats is addressed again and here also a reference is given (line 54).

  1. R2: Result: To improve results in other trials of this project, I suggest including two new control groups. Using a similar treatment strategy, but repeating eprinomectin (twice) and moxidectin (twice). However, I think this is no reason to reject this article, with a short communication format.

Comment: Thank you for your consideration. We are aware about the fact that the necessity of a control group for FECRT is intensively discussed amongst experts and thus we are waiting for the recommendations that will soon be given in the new WAAVP guidelines. However, in Kaplan et al. (Kaplan RM. 2020. Biology, Epidemiology, Diagnosis, and Management of Anthelmintic Resistance in Gastrointestinal Nematodes of Livestock. Veterinary clinics of North America: Food Animal Practice, 36, 17-30.) it is already indicated that control groups may be omitted in the new WAAVP guidelines.  Therefore, we decided to already not include control groups. Furthermore, in our case all animals with high EpG required treatment.

  1. R2: Methodology: Add the age of treated animals

Comment: The age is given in the supplementary table 1